# Ultrafine Particles Issued from Gasoline-Fuels and Biofuel Surrogates Combustion: A Comparative Study of the Physicochemical and *In Vitro* Toxicological Effects

**DOI:** 10.3390/toxics11010021

**Published:** 2022-12-26

**Authors:** Ana Teresa Juárez-Facio, Tiphaine Rogez-Florent, Clémence Méausoone, Clément Castilla, Mélanie Mignot, Christine Devouge-Boyer, Hélène Lavanant, Carlos Afonso, Christophe Morin, Nadine Merlet-Machour, Laurence Chevalier, François-Xavier Ouf, Cécile Corbière, Jérôme Yon, Jean-Marie Vaugeois, Christelle Monteil

**Affiliations:** 1Univ Rouen Normandie, UNICAEN, ABTE UR 4651 F, 76000 Rouen, France; 2Univ Rouen Normandie, INSA Rouen, CNRS, COBRA, 76000 Rouen, France; 3Univ Rouen Normandie, INSA Rouen, CNRS, GPM-UMR6634, 76000 Rouen, France; 4Institut de Radioprotection et de Sureté Nucléaire, PSN-RES, SCA, LPMA, 91192 Gif-sur-Yvette, France; 5Univ Rouen Normandie, INSA Rouen, CNRS, CORIA, 76000 Rouen, France

**Keywords:** *in vitro*, BEAS-2B, biofuels exhaust, ultrafine particles, air-liquid interface exposure, physicochemical characterization, DTT assay

## Abstract

Gasoline emissions contain high levels of pollutants, including particulate matter (PM), which are associated with several health outcomes. Moreover, due to the depletion of fossil fuels, biofuels represent an attractive alternative, particularly second-generation biofuels (B2G) derived from lignocellulosic biomass. Unfortunately, compared to the abundant literature on diesel and gasoline emissions, relatively few studies are devoted to alternative fuels and their health effects. This study aimed to compare the adverse effects of gasoline and B2G emissions on human bronchial epithelial cells. We characterized the emissions generated by propane combustion (CAST1), gasoline Surrogate, and B2G consisting of Surrogate blended with anisole (10%) (S+10A) or ethanol (10%) (S+10E). To study the cellular effects, BEAS-2B cells were cultured at air-liquid interface for seven days and exposed to different emissions. Cell viability, oxidative stress, inflammation, and xenobiotic metabolism were measured. mRNA expression analysis was significantly modified by the Surrogate S+10A and S+10E emissions, especially *CYP1A1* and *CYP1B1*. Inflammation markers, *IL-6* and *IL-8*, were mainly downregulated doubtless due to the PAHs content on PM. Overall, these results demonstrated that ultrafine particles generated from biofuels Surrogates had a toxic effect at least similar to that observed with a gasoline substitute (Surrogate), involving probably different toxicity pathways.

## 1. Introduction

Epidemiological and experimental studies have reported compelling evidence that pollution by airborne particulate matter (PM) is an important cause of health outcomes associated with respiratory and cardiovascular diseases as well as cancer mortality [1,2,3,4]. In urban areas, the traffic sector is a major contributor to PM emissions as well as other pollutants, including carbon monoxide (CO), volatile organic compounds (VOCs), nitrogen oxides (NOX), polycyclic aromatic hydrocarbons (PAHs), and PM [5,6]. Moreover, diesel exhaust has been classified as carcinogenic (Group 1) and gasoline exhaust as possibly carcinogenic (Group 2B) to humans by the International Agency for Research on Cancer (IARC) [7].

Today’s transports depend mainly on fossil fuels, and alternatives are needed against increasingly stringent emission regulations, rising oil prices, and a finite supply of fossil fuels. Biofuels are renewable, sustainable, efficient, and cost-effective energy sources [8] and represent an attractive option to replace fossil fuels. Biofuels blended with gasoline have been gradually introduced to the global market; an example is E10, constituted of 10% ethanol and 90% gasoline. Among the diversity of biofuels, second-generation biofuels (B2G) are derived from lignocellulosic biomass. The special feature of the B2G is that “nonfood biomass” such as agricultural waste, straw, grass, and wood, is used as raw materials for their production [9,10,11]. Compared to the abundant literature on diesel and gasoline emissions, relatively few studies focus on B2G and their health effects. Several studies have shown that adding ethanol to gasoline reduces carbonaceous pollutants, such as carbon monoxide (CO), total hydrocarbons (THC), and particulate matter. However, some studies also reported the formation of more complex compounds, such as acetaldehydes, that could lead to greater health effects [12,13,14].

The mechanisms emphasizing the health outcomes of PM are considered to involve mainly oxidative stress and inflammation. Oxidative stress results from an imbalance between the generation of reactive oxygen species (ROS) and the antioxidant defense. ROS are indispensable for regulating critical signaling pathways involved in cell growth, proliferation, differentiation, and survival. However, an excess of ROS can lead to oxidative stress [15]. Hence, the measurement of the oxidative potential (OP) of PM could be used as an indicator of particle toxicity. The OP can be defined as the PM capacity to oxidize target molecules and induce the formation of oxidizing species in the lung [16]. This evaluation can be performed by various acellular methods, such as the DTT assay (OP_DTT_) or the antioxidant depletion assay (OP_AO_), which are simpler and quicker to achieve than cellular assays. Moreover, studies have established a link between OP and biological responses to particle exposure, making it a relevant toxicological metric [17,18,19].

Besides chemical composition, size is also an essential factor for PM assessment. Fine particles (FP) and ultrafine particles (UFPs) have attracted more attention as they can enter deeper into the respiratory tract [20]. UFPs are also associated with lower mass concentrations than coarser fractions; they are more numerous and have a higher specific area, which may significantly impact their consequences on human health [4,21]. The health effects of traffic emissions from diesel and gasoline engines have been extensively studied in vitro; nevertheless, there are few studies concerning new fuel emissions, and in most of the studies, particle suspensions were used for submerged exposures [22,23,24,25]. Therefore, new approaches to assessing health effects from particles, especially UFPs, are needed to support biofuel development, but generating realistic aerosols and performing in vitro exposures at relevant particle-doses is challenging.

The aim of this study was to compare the physicochemical and toxicological characteristics of UFPs emitted during the combustion of gasoline fuel and biofuel Surrogates under controlled combustion and exposure conditions. This study focused on one variable, the fuel: propane as a reference fuel, a gasoline surrogate, and two biofuel surrogates consisting of a Surrogate blended with anisole to mimic a B2G or ethanol to mimic an E10. These Surrogates were combusted under identical operating conditions, and the physicochemical characterization was performed in order to investigate how the composition of these four types of UFPs affects their biological effects *in vitro*.

## 2. Materials and Methods

### 2.1. Chemicals

Unless otherwise specified, chemicals were purchased from Sigma-Aldrich (St. Quentin Fallavier, France).

### 2.2. Exposure System and Operating Conditions

The exposure system used has been previously described [26]. In this study, the miniCAST (Jing Ltd., Zollikofen, Switzerland) soot generator, initially designed for gas fuels, was modified to burn both propane and liquid fuels (Appendix A). In gas mode, a propane flow is mixed with nitrogen, and oxidation air flows to produce a stable flame, while in liquid mode, a syringe pump and an evaporator (CEM-W-102A) replace the propane flow. The syringe pump controls fuel flow towards the evaporator, which atomizes, evaporates, and transports fuel vapors through a nitrogen flow. In this study, a miniCAST soot generator was used with propane (CAST1 condition) or liquid fuels consisting of Surrogate gasoline fuel and a Surrogate B2G fuel.

Gasoline comprises many chemical species, including alkanes, alkenes, and aromatic compounds. Moreover, fuel compositions may vary between petrol stations and storage conditions. The Surrogate gasoline composition used in this work was the same as Wu et al. 2017, which modeled a commercial gasoline fuel with the main chemical families [11]. In order to model a B2G, we used two additives: ethanol and anisole. Ethanol (10 v%) was selected as it is currently on the global market under different blend ratios ranging from 5% to 85%. Anisole (10 v%) was chosen as a second additive to the Surrogate as it represents rich phenol species in Surrogates during combustion. Indeed, it was reported that lignocellulosic biomass is transformed into a mixture of rich phenol species [27]; hence, anisole represents an optimal additive to model B2G. Surrogate gasoline fuel (S) consisted of hexane (24.31 wt%), 2,3-dimethyl-2-butene (8.15 wt%), cyclohexane (14.21 wt%), isooctane (17.75 wt%), and toluene (35.58 wt%) in accordance to [11]. Table 1 shows the flow rate of the operating conditions in this study.

Aerosols from the miniCAST were diluted at 50 mg/m^3^ by adding airflow before being delivered to cells through the Vitrocell^®^ exposure system (Vitrocell^®^ Systems GmbH, Waldkirch, Germany). In this way, the same exposure dose was expected for each aerosol.

### 2.3. Characteristics of Aerosols Emission

#### 2.3.1. Particles Characterization: Size, Mass, and Morphology

Number and particle size distributions were measured with a Scanning Mobility Particle Sizer (SMPS) (TSI, MN, USA), and mass concentration was determined by Tapered Element Oscillating Microbalance (TEOM) (Rupprecht & Patashnick Co., Inc., Albany, NY, USA) measures. For electron microscopy investigations, TEM grids were positioned inside the Vitrocell^®^ exposure system, replacing the epithelial cell line insert, to collect soot particles under the same conditions as for ALI. TEM images were acquired using a JEOL 2010F (JEOL, Tokyo, Japan) at 120 KV in bright field conditions. Soot images were analyzed following the method proposed by Bescond et al. 2014 to determine primary particle size [28]. Particles were sampled on cellulose filters for the untargeted analysis and on quartz filters for the targeted analysis and OC/TC measurements. The OC/TC ratio was determined through thermo-optical analysis (Sunset Lab, IMPROVE protocol).

#### 2.3.2. Chemical Characterization

The methods to analyze particle and gaseous phases are the same as those described in previous works [29]. Briefly, samples were collected in Tedlar bags for gaseous emissions and then analyzed by gas chromatography with a thermal conductivity detector (GC/TCD) (SCION Instruments UK Ltd., Scotland, UK) and gas chromatography coupled to a mass spectrometer (GC/MS) (PerkinElmer, MA, USA). The untargeted analyses from CAST1 and Surrogate PM were performed using laser desorption ionization (LDI) with ultra-high-resolution Fourier transform ion cyclotron resonance mass spectrometry (FTICR MS). The particulate matter samples were deposited on an MTP 384 ground steel target plate using a solvent-free method described elsewhere [30,31]. Analyses were performed with a SolariX XR FTMS (Bruker Daltonics, Billerica, MA, USA) FTICR with a 12 T superconductive magnet. An Nd-YAGx3 laser emitting a 355 nm photon at a frequency of 1000 Hz was used for the LDI source, which operated in positive ion mode. A laser power from 17% to 18% was used, 50 laser shots were summed per scan, and a total of 30 scans were accumulated between *m*/*z* 110.6 and *m*/*z* 1200. The data came as 1.26 s transients sampled with 4 M words. Ion transmission and analyzer parameters can be found in Appendix A. Triplicate analyses of the sample were performed. Internal mass calibration (with a quadratic function) was performed using a list of PAHs found in Appendix A, resulting in a standard deviation of around 50 ppb. Mass spectrometry signals were attributed to molecular formulas using Bruker DataAnalysis v5.1 (Bruker Daltonics) with an error tolerance of 0.5 ppm and molecular formula boundaries C_c_H_h_N_0–1_O_0–3_. The data from the triplicate analysis were merged into one list where all attributions were considered, and attributions not found within a sample were set to zero intensity. The elemental composition attributions of the average spectra were used for graphical representations and compared via multivariate data analysis. Principal Component Analyses (PCA) were performed using a Matlab lab-written script. Data were subjected to power transformation before PCA, to reduce heteroscedasticity and to do pseudo-scaling. The resulting scores and loadings data, combined with the elemental composition attribution, were used for visualization. In this respect, score data are linked to the experiments (replicates and sample type) and loading data to the individual elemental compositions (molecular profile).

### 2.4. Oxidative Potential

#### 2.4.1. Particle Suspension Preparation

Particles were collected on cellulose filters placed after the dilution column. Particles were removed from the filters by scratching, were suspended in ultrapure water at 500 µg/mL, and sonicated for 60 min at 40 kHz. As described in our previous studies, diesel particles were used as a positive control for oxidative potential measurement [32,33]. Then, particle suspensions were aliquoted and stored at −20 °C.

#### 2.4.2. Dithiothreitol (DTT) Assay

The Dithiothreitol assay monitors the oxidation of DTT, a reducing agent, in the presence of particles. First, particles were incubated, in a 96-well plate, with 1 mM DTT at 37 °C. Seven concentrations of particles were tested from 0.56 to 56 µg/mL. After 1 h of incubation, the plate was centrifuged at 2100× *g* and 4 °C for 15 min. Then, 50 µL of supernatants were recovered and added to an equal volume of 5,5′-dithiobis-(2-nitrobenzoic acid) (DTNB) (2.5 mM). Then, the remaining reduced DTT reacted with DTNB to produce 2-Nitro-5-thiobenzoic acid (NTB), a yellow product that absorbs at 412 nm. The absorbance was measured with a plate reader (Safas Xenius, Monaco), and the percentage of depletion was calculated according to the following equation [18]:%DTT depletion = 100 − (DTT concentration in the sample/DTT concentration in the blank) × 100(1)

#### 2.4.3. Antioxidant (AO) Depletion Measurement

This second acellular assay measures the depletion of ascorbic acid (AA). Briefly, a range of particle concentrations from 0.56 to 56 µg/mL was prepared in a synthetic respiratory tract lining fluid (RTLF) containing 0.9% NaCl at pH 7.4 and AA (200 µM). Solutions were incubated for 4 h at 37 °C, then centrifuged for 10 min at 15,000 rpm and 4 °C. An aliquot of each supernatant was diluted 1:5 in the mobile phase (Potassium hydrogen phosphate, 10 mM, pH 3). Solutions were filtered before the injection with a Nylon 0.45 µM syringe filter (Cloup, Champigny-sur-Marne, France). Finally, samples were injected into an Agilent High-Pressure Liquid Chromatography system 1260 Infinity II, equipped with a DAD detector (Agilent Technologies, Santa Clara, CA, USA). The chromatography was equipped with a Poroshell 120 EC-C18 (3 × 150 mm, 2.7 µm) with a guard EC-C18 (3 mm) maintained at 25 °C. The flow rate of the mobile phase (100%) was 0.6 mL/min. UV data was collected at 220 nm. This method was validated to meet the ICH requirements. The percentage of AO depletion was calculated according to the following equation [18]:% AO depletion = 100 − (AO concentration in the sample/AO concentration in blank) × 100(2)

### 2.5. Cell Culture and Exposures

The bronchial epithelial BEAS-2B cell line (ATCC—CRL9609TM) was cultivated in LHC-9 medium (GIBCO) on a collagen-coated TranswellTM insert as described before [28]. Culture at the liquid-liquid interface was maintained for 10 days before aerosol exposure, performed at the air-liquid interface (ALI) under standard airway epithelial cell growth conditions (37 °C, 95% humidity, and 5% CO_2_). All exposures were performed using BEAS-2B cells between passages 40 and 50.

Cells were exposed to fresh aerosols for 35 min with an electrostatic field (+1000 V), while control samples were exposed to ambient HEPA-filtered laboratory air through the Vitrocell System. The aerosol was set at 50 mg/m^3^ to allow a similar deposition in all conditions, corresponding to an estimated exposure dose of 370 ng/cm^2^. For each fuel, at least four independent exposure experiments were performed. The response of particle-exposed cells was compared to that of the control cells 3 h and 24 h after exposure.

### 2.6. Biological Endpoints

#### 2.6.1. Cytotoxicity

The MTT assay is a common method to assess cell viability. The protocol was previously described [29]. Briefly, 3 h and 24 h after exposure, cells were incubated with MTT solution for 3 h, and then formazan crystals were dissolved with SDS solution (10%). The absorbance was measured at 570 nm (Xenius, Safas, Monaco) and normalized to the control, set at 100%.

#### 2.6.2. ATP/ADP Quantification

##### Cell Sample Preparation

ATP and ADP levels were determined on perchloric extracts as previously described [34]. Briefly, after 3 h and 24 h exposure, inserts were gently rinsed with PBS-1X. Then, 500 µL of perchloric acid (1 N) was added to each insert to harvest cells. The samples were centrifuged at 13,500× *g* for 5 min at 4 °C. Pellets were resuspended in NaOH (1 N) and used to determine protein content by the Lowry assay. The supernatant was neutralized by the addition of K_2_CO_3_ and then samples were centrifuged again. Supernatants were stored at −80 °C until measurements.

##### ATP/ADP Quantification

ATP and ADP quantification were performed with the Agilent High-Pressure Liquid Chromatography system 1260 Infinity II equipped with a DAD detector (Agilent Technologies, Santa Clara, CA, USA). Stock solutions of 10 mM ATP (adenosine 5′-triphosphate disodium salt hydrate > 99%), ADP (adenosine 5′-diphosphate sodium salt > 98%), and AMP were prepared by dissolving the appropriate amounts in ultrapure water. The above stock solutions were stored at −20 °C, and the working solutions were prepared daily by appropriate dilution in the mobile phase. The mobile phase consisted of 50 mM potassium hydrogen phosphate (pH 6.80). Analytes were separated on a Poroshell 120 EC-C18 (3 × 150 mm, 2.7 µm) with a guard EC-C18 (3 mm) at 254 nm. The chromatographic separation was carried out at 20 °C with 0.6 mL/min as the flow rate. The identification of peaks was based on the comparison of retention times and diode-array spectra of chemical standards and biological samples. The analytical method was validated by assessing the linearity, limits of quantification, and detection and accuracy to meet the ICH requirements.

#### 2.6.3. qRT-PCR

A gene expression study was performed as previously described [29]. Total ARN was extracted from cells using TRI-REAGENT^®^ and Direct-zol™ RNA MiniPrep (Ozyme). The RNA concentration and purity were measured with a NanoDrop spectrophotometer (NanoDrop™ 2000/2000c.). Treatment for DNase was performed with DNase I, Amplification Grade (InvitrogenTM). cDNA was prepared by reverse transcribing extracted RNA samples using Invitrogen™ Transcriptase inverse (M-MLV). The expression of mRNA was determined by the quantitative real-time polymerase chain reaction (qRT-PCR) using Brilliant III Ultra-fast SYBR Green QPCR Master Mix (Agilent Technologies). Expression profiles of *NQO1*, *HO-1*, *IL-8*, *IL-6*, *CYP1B1*, and *CYP1A1*, were normalized to B2M. Results were expressed as relative expression using the 2^−ΔΔCT^ method [35].

### 2.7. Statistical Analysis

Biological endpoint results were expressed as mean ± SD. Statistical analysis was performed using the non-parametric Mann–Whitney U-test (GraphPad for Windows, v9.0, San Diego, CA, USA). Statistically significant differences were reported with *p* < 0.05.

## 3. Results

### 3.1. Aerosol Characterization

#### 3.1.1. Particles’ Characterization: Size, Mass, and Morphology

The size distribution and morphological parameters of the studied aerosols are reported in Table 2. The studied aerosols presented very close geometric median mobility diameter (D_m,geo_) values (114–123 nm). Liquid fuels show a bigger size distribution of the primary particles (D_p,geo_) (Surrogate: 43.3 nm, S+10A: 35.5 nm, S+10E: 34.9 nm) compared to the propane condition (CAST1: 26.6 nm) and the addition of ethanol or anisole to the Surrogate reduces this value by respectively 19% and 18%. The presence of oxygenated compounds is suspected to play a role, certainly by promoting the oxidation process. Similar amounts of organic compounds were presented (% OC/TC), CAST1: 4.1 ± 3.5%, Surrogate: 3.5 ± 0.4%, S+10A: 5 ± 0.6%, and S+10E: 7.7 ± 0.9.

Some images obtained by transmission electron microscopy are reported in Figure 1 for the different studied fuels. All images report classical soot morphologies, i.e., fractal aggregates made of slightly overlapped primary spheres. In that sense, the selected images are representative. Concerning the size of the primary spheres, a statistical analysis provided the median diameter and corresponding geometric standard deviation reported in Table 2. The fractal dimension that quantifies the aggregate compacity is not reported here but is representative of classical soot particles (around 1.7).

#### 3.1.2. Chemical Characterization

Particle and gaseous phases from aerosol emission of different fuels combustion were characterized. For the gaseous phase, data are specified in molar percentage, and precise values are available in Appendix A. It was noticed that CAST1 generated more volatile compounds (1% molar), with carbon dioxide being the main compound. In the studied conditions, carbon dioxide was found at 0.6%. Hydrogen was absent from CAST1, at 0.03% for the Surrogate, S+10A, and S+10E. The same trend was noticed for methane, not found in CAST1, at around 0.03% for Surrogate, S+10A, and S+10E. C2 hydrocarbons (ethylene, acetylene, and ethane) were absent from CAST1 and, for ethane, around 0.0007% for Surrogate, S+10A, and S+10E. Acetylene was around 0.01% in Surrogate and S+10A, and at 0.013% for S+10E. Ethylene was found at 0.016% in Surrogate, 0.018% in S+10A, and 0.023% in S+10E. In CAST1, only propane was detected at 0.0005%, without other C3 hydrocarbons. An amount of 0.004% was noticed for propene in Surrogate, S+10A, and S+10E, 0.001% for propadiene in Surrogate, S+10A, and S+10E. An amount of 0.002% was found for propyne in Surrogate, S+10A and S+10E. C4 and C5 hydrocarbons were absent from CAST1 and around 0.001% for isobutene, 0.0007% for 1-buten-3-yne and 0.002% for 1.3 butadiene in surrogate, S+10A and S+10E. A particular case was noted for cyclopentadiene, which was found much more when 10% anisole or 10% ethanol was added to the Surrogate (0.0023% for S+10A and 0.017% for S+10E compared to 0.0011% for Surrogate). Finally, benzene and toluene were not found in CAST1 and around 0.00004% molar in Surrogate, S+10A, and S+10E.

The targeted analysis tested the 16 priority PAHs and 5 oxy PAHs (Appendix A). With the CAST1 condition, the predominant PAHs were acenaphthylene, phenanthrene, fluoranthene, and pyrene, as previously published [29]. These PAHs represent around 70% of the total amount. The particles obtained from the gasoline Surrogate combustion present a similar PAHs profile, with a predominance of phenanthrene (23% of the total amount) and acenaphthylene (25% of the total amount). The latter represents more than 80% of the total PAHs quantified in particles obtained from the S+10A condition. The oxy-PAHs amounts represent 2.8% for the S+10A condition, 4.5% for CAST1, and 8% for the Surrogate.

Untargeted analysis was performed in triplicate for CAST1 and Surrogate fuels PM. Graphical representations and principal component analysis (PCA) obtained from the list of molecular formulas from all mass spectra are shown in Figure 2. The relative intensity by molecular classes, presented in Figure 2a for the four analyzed samples, showed slight differences between combustion conditions and fuels. For all fuels and conditions, the ion relative intensity was dominated by the hydrocarbon class (HC) (around 80%), followed by compounds with one and two oxygen atoms (around 17% and 3%, respectively). However, CAST1 showed a higher relative intensity for the hydrocarbon molecular class and a lower relative intensity for the oxygenated molecular classes.

The differences between CAST1 and the Surrogate fuels were better revealed on the PCA score plot in Figure 2b was constructed from the table of identified molecular formulas and associated ion intensities. The first principal component (PC1) represented 71% of the total variation and efficiently separated the CAST1 condition from the Surrogate conditions, highlighting significant differences in the chemical composition of the PM in CAST1 compared to the combustion of Surrogate fuels. The second principal component (PC2), accounting for 9% of the total variation, showed that the Surrogate conditions, on the other hand, yielded very similar compounds.

The coefficients obtained from the principal component analysis were used in the graphical representation of the double bond equivalent (DBE) vs. carbon number diagram, as shown in Figure 2c,d DBE, also called the “degree of unsaturation,” totals the number of rings and π bonds in a molecule. For ions, DBE values can be either integer or half-integer numbers depending on the type of ion: M+ (with an odd number of electrons) or [M + H]+/[M − H]+ (with an even number of electrons). Here, only M+• ions are shown as they are the privileged ion form of polycyclic aromatic hydrocarbons (PAHs). The three lines represented in Figure 2c,d correspond to the cata-condensed PAHs (in red), peri-condensed PAHs (in blue), and elemental carbon (in black) structures as described in various studies [37,38,39,40]. The cata-condensed PAHs do not present any carbon shared with more than two aromatic rings, whereas peri-condensed structures show carbon atoms shared with more than two aromatic rings.

The DBE vs. carbon number diagrams of hydrocarbon (HC) and hydrocarbon with one oxygen atom (CHO_1_) molecular classes showed a main distribution of ions from 15 carbon atoms to 70 carbon atoms with a DBE value from 10 to 60. Nearly all the ions were between the cata-condensed and peri-condensed lines, indicating highly aromatic compounds were evidenced. From the PC1 coefficients associated with each molecular formula, which are used for color coding of each ion for the HC class, we observed that aromatic compounds with lower masses were characteristic of the CAST1 samples, while Surrogate fuels could be distinguished by a broader distribution of molecular formulas that also included hydrocarbon compounds with a higher number of carbon atoms and thus higher masses.

For the CHO_1_ class, the negative PC1 coefficient associated with the ions showed that O-HAP was more abundant in Surrogate samples, which is consistent with the presence of oxygen atoms in the composition of several molecules in the Surrogate fuels.

### 3.2. Oxidative Potential (AO Depletion Measurement and DTT Assay)

Two acellular assays studied the oxidative potential of particles: the dithiothreitol assay and antioxidants depletion measurement (Figure 3). Several concentrations of PM were tested, from 0.56 to 56 µg/mL. The lowest concentration corresponds to the exposure dose in the ALI system. First, a dose-dependent of DTT consumption was observed for CAST1. CAST1 particles induce the highest DTT loss rate, with an approximately 60% loss rate at the highest dose. Concerning Surrogate, S+10A, and S+10E particles, DTT loss rate values at 56 µg/mL were lower, around 20%. Then, the depletion of ascorbic acid (AA) was induced by CAST1 particles for all tested concentrations. Lower AA oxidation was observed for Surrogate and S+10E particles, but only at concentrations from 2.8 µg/mL to 56 µg/mL, respectively. Moreover, the AA depletion percentage in S+10A particles did not exceed 6% at the highest concentration tested.

### 3.3. Cell Response after Exposure

BEAS-2B cells were exposed to CAST1, Surrogate, S+10A, and S+10E emissions for 35 min, and biological endpoints were performed after 3 h and 24 h of incubation with particles. Results from MTT assays presented in Figure 4 showed a significant reduction in cell viability compared to the air-exposed cells (control) after 3 h of incubation for the CAST1, S+10A, and S+10E conditions. However, after 24 h of incubation, only the Surrogate and S+10E conditions induce a decrease in cell viability.

The effects of emission exposures on ATP and ADP levels compared with air-exposed cells (control) are shown in Table 3. Globally, the levels of ATP remained stable compared to controls. The only significant effect observed was a diminution in the ADP levels after 24 h of incubation with S+10A particles compared to the control (*p* < 0.05).

Figure 5 shows the ATP/ADP ratios normalized with the corresponding control and calculated from Table 3. Compared to air-exposed cells, the particles generated from Surrogate fuel increased this ratio after 3 h and decreased it after 24 h compared to 3 h. After exposure to particles generated from S+10A, this ratio slightly decreased after 3 h.

As the MTT assay indicated mild cytotoxicity (<20%) and ATP measures showed significant differences only for Surrogate particles, further evaluation was performed to identify the potential biological markers that aerosols may alter. Oxidative stress, inflammation, and xenobiotic metabolism markers were studied by qRT-PCR. Results in Figure 6 show several up- and down-regulations in all markers at both times after exposure compared to air control. First, *NQO1* mRNA was significantly induced by particles generated from the S+10A at 3 h and the Surrogate at 24 h (Figure 6A). *HO-1* mRNA was significantly up-regulated by the Surrogate and S+10A particles at 3 h (Figure 6B). At 24 h, only the Surrogate particles reported a significant induction of *HO-1* mRNA. CAST1 and S+10E mRNA showed no modification at both time points of incubation. Next, the inflammation marker IL-6 expression was reduced by CAST1 and S+10E at 3 h, and this response was kept at 24 h only for the S+10E. No changes were observed for the Surrogate and S+10E (Figure 6C). Then, inflammation marker *IL-8* expression was significantly reduced by CAST1 and S+10A particles at 3 h, while at 24 h the Surrogate and S+10A were increased and S+10E was significantly reduced (Figure 6D).

Finally, two markers of xenobiotic metabolism were studied. *CYP1A1* and *CYP1B*1 mRNA expressions were up-regulated by both Surrogate, S+10A and S+10E particles after 3 and 24 h exposures, whereas the CAST1 particles showed a down-regulation for *CYP1B1* after 3 h exposure (Figure 6E,F).

## 4. Discussion

The use of biofuels results in multiple technical, economic, environmental, and health challenges. In terms of health, the main interest is the reduction of harmful emissions. The aim of this study was to compare the physicochemical profile and the toxicological effects of gasoline and B2G fuels.

Gas analysis showed that for carbon dioxide, the highest levels were by the CAST1 condition, and the addition of anisole to the Surrogate slightly increased the amount while the ethanol slightly reduced it. Regarding the chemical composition of PM, PAHs were identified as major contributors, especially from liquid fuels, compared to CAST1 propane. The addition of anisole to the Surrogate is the condition showing the higher PAHs amounts, mainly acenaphthylene and naphthalene. This agrees with several studies showing that biofuel combustion, especially from ethanol blends, reduces carbon dioxide, hydrocarbons, PM, and PAHs emissions while increasing NOx and aldehydes [41,42,43,44]. The particle size measurements showed that the addition of an oxygenated additive (ethanol or anisole) decreased the geometric diameter of the primary spherules constituting soot aggregates, as previously observed by Verma et al., 2021, from biodiesels. However, the CAST1 primary particles presented the smallest size [45].

The smaller-sized particles could result in higher oxidation reactivity given their larger specific surface area. This can be detected by measures of oxidative potential (OP), an indicator of the biological reactivity of particles. Two types of acellular tests were used in this work, using increasing concentrations of particles. Results demonstrated that both DTT and ascorbic acid were more depleted by CAST1 particles compared to the other particles. This observation agrees with the hypothesis that size may contribute to OP, given a larger specific area. However, for the smallest concentration, OP^DTT^ results demonstrated that particles from liquid fuels depleted more DTT than CAST1 particles, probably due to PAHs, which are known to correlate strongly with DTT [46,47]. However, this depletion remained modest, contrary to the results obtained with the CAST1 particles. The OP^AA^ results were less consistent, probably due to the absence of metals (i.e., Mn, Fe, Cu, Cr, and Zn) known to drive the AA response [48]. Indeed, important contents of Fe, Cu, and Zn are found in emissions from commercial diesel and gasoline fuels [17,47,49]. Hence, further analyses are needed to elucidate the complexity of metal content in fuel emissions and their involvement in the oxidative potential of emitted UFPs.

As inhaled UFPs deposition may occur at the bronchial level [20], the bronchial epithelial BEAS-2B cells were used in this study. BEAS-2B cells were exposed to freshly generated aerosols from the miniCAST, and biological endpoints were performed after 3 or 24 h of incubation with particles. Each aerosol was diluted to 50 mg/m^3^ for a comparable estimated depositional dose of particles (370 ng/cm^2^) in order to investigate an early and a late response. The cytotoxicity performed by the MTT assay was reported to be <15%, reflecting mild cytotoxicity after 3 h and 24 h, associated with alterations in the ATP/ADP ratio that did not persist at 24 h except for the Surrogate. The ATP/ADP ratio is a critical parameter of cellular energy status that regulates many metabolic pathways [50]. This ratio is an indicator of the energetic cellular state, which may reflect an increase in the hydrolysis of ATP and, therefore, its consumption or a decrease in its production. Although these effects on energetic metabolism were modest, it can be assumed that they could have cellular consequences if prolonged or repeated exposures occurred. Indeed, the reduction of ATP affects many essential cellular systems, such as motor protein functions, including ciliary motility and intracellular transport. Although further studies are needed to explain the origin of the alterations observed here and their potential consequences, these results are consistent with previous studies, highlighting impaired energy homeostasis after particle exposure. A previous study has shown that airborne particles may disrupt the alveolar/endothelial barrier function, which is tightly regulated by intracellular ATP [51]. Jin et al. have also shown an alteration of ATP production and energy metabolism in the lungs of rats after sub-chronic PM_2.5_ exposures [52]. These observations outline the use of ATP assessments as an indicative biomarker of PM exposures.

Our results on *NQO1* and *HO-1* mRNA expression did not show important effects regarding oxidative stress. *NQO1* reported a slight up-regulation by the S+10A condition at 3 h and by Surrogate at 24 h. *NQO1* is induced by Nrf2 and exerts cytoprotective, antioxidant, and anti-inflammatory effects in the lungs [53] against particles, which leads to its induction [22,54]. Concerning *HO-1*, Surrogate and S+10A induced an important up-regulation 3 h after exposure, and after 24 h, expression was restored to normal values. This kinetic of HO-1 expression response is similar to that observed in a previous study with organic ultrafine particles [29], underlining the cellular adaptation to transient production of reactive oxygen species through the induction of antioxidant response. Interestingly, this result agrees with the results of the OP at the lowest dose, which represents the same order of magnitude as that estimated in culture (more precisely 1.5 times). Inflammation response and, more specifically, *IL-8* and *IL-6* mRNA expression were also studied by qRT-PCR. *IL-8* was slightly downregulated at 3 h, but the opposite response was observed at 24 h. IL-6 showed similar results. Studies on PFs and UFPs mainly reported the induction of inflammation genes at higher exposure doses [55,56] as well as cytokine release after longer incubation times [57,58]. In addition, immunosuppressive effects have been seen to be related to PAHs amount [59]. Our results suggest that a down-regulation of *IL-8* and *IL-6* may be associated with an early inflammation response, probably due to the organic content. Further research should consider measuring cytokine release to validate these conclusions. The xenobiotic metabolism was also considerably disturbed. *CYP1A1* and *CYP1B1* mRNA were highly upregulated by liquid fuels at both time conditions. These two cytochrome P450 (CYP) enzymes may be induced by PAHs presented in particles. According to the above-reported results, CAST1 does not significantly impact the gene expression of the studied markers, while Surrogate and S+10A or S+10E do.

Despite the interesting approach proposed in this study, there are a few limitations. Although using a soot generator and Surrogate fuels allows the generation of reproducible aerosol emissions, in real-world conditions, fuel emissions encounter physicochemical processes such as oxidation and interaction with other pollutants that can modify PM characteristics, resulting in different effects compared to the laboratory conditions. Indeed, the composition of the exhaust particles can vary according to parameters such as the type and age of the engine, the drive cycle, the presence of additives in fuels, etc. Another limitation of our work concerns the in vitro model based on a cell line. The BEAS-2B cell model is widely used in particle toxicology; nevertheless, primary cell models are more representative of the pulmonary epithelium.

## 5. Conclusions

This study demonstrated that ultrafine particles generated from a gasoline substitute mixed with anisole or ethanol had a toxic effect at least similar to that observed with a gasoline substitute, involving probably different toxicity pathways. Although these ultrafine particles did not present an important oxidative potential, they altered the expression of xenobiotic metabolism-related genes in BEAS-2B cells. Further studies are required to study the underlying molecular mechanisms involved in these toxicity pathways and their consequences on health. The integrated approach used in this work represents a valuable methodology to test novel fuel emissions, linking chemical characterization and toxic response, and can be appropriate to test B2G before commercialization.

## Figures and Tables

**Figure 1 toxics-11-00021-f001:**
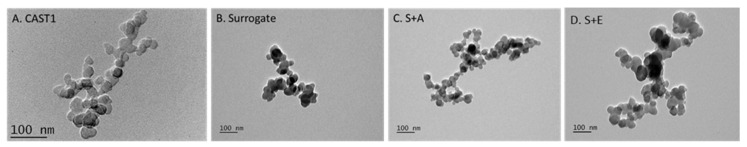
Transmission electron microscopy images of particles from: (**A**) CAST1; (**B**) gasoline Surrogate; (**C**) B2G Surrogate anisole (S+10A); and (**D**) Surrogate ethanol (S+10E).

**Figure 2 toxics-11-00021-f002:**
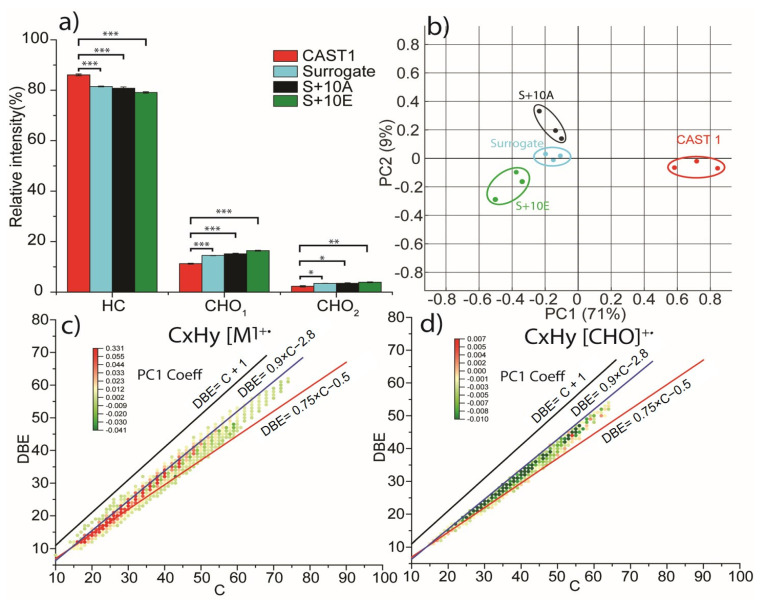
(**a**), Ion relative intensity for hydrocarbons and compounds with carbon, hydrogen, and one or two oxygen atoms (CHO_1–2_) for CAST1 and Surrogate fuels (n = 3). *T*-test *p*-values are reported —between CAST 1 and Surrogate fuels as: * for *p* < 0.05, ** for *p* < 0.01, *** for *p* < 0.001; (**b**) principal component analysis score plot from the list of intensities of molecular formulas c), double bond equivalent (DBE) vs. carbon number diagram for hydrocarbon radical ions color-coded using PC1 coefficient and d), double bond equivalent vs. carbon number for radical ions with one oxygen atom color-coded using PC1 coefficient. For both diagrams (**c**,**d**), the lines represent the trend of DBE vs. carbon number for the cata-condensed polycyclic aromatic hydrocarbons (PAHs) (in red), peri-condensed PAHs (in blue), and elemental carbon (in black) structures.

**Figure 3 toxics-11-00021-f003:**
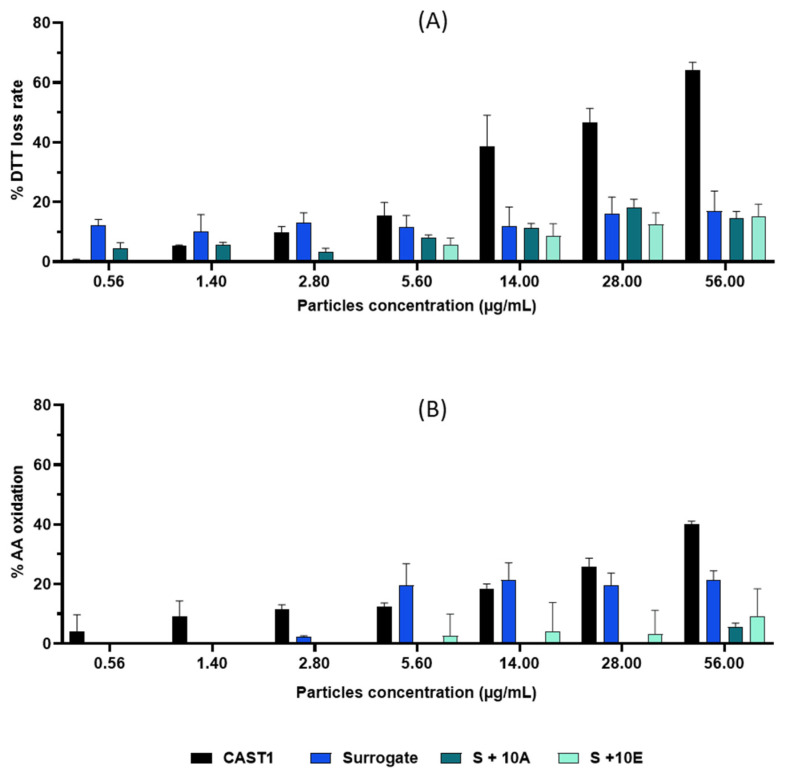
Oxidative potential of particles from CAST1, Surrogate, S+10A, and S+10E by DTT assay (**A**) and AA depletion measurement (**B**).

**Figure 4 toxics-11-00021-f004:**
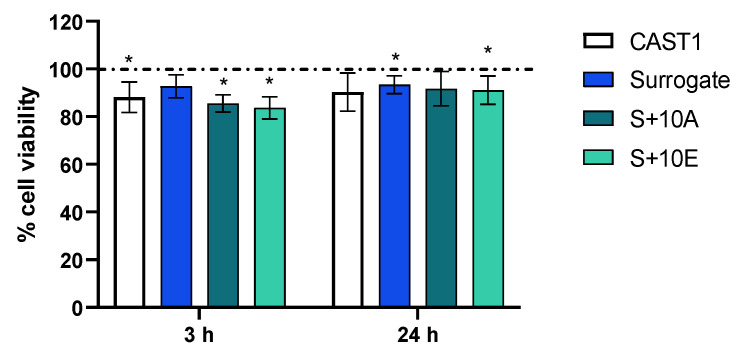
Cytotoxicity determined by MTT assays after 3 h and 24 h of incubation after aerosol exposure. Data are expressed as mean ± SD, n = 4–5 independent experiments, * *p* < 0.05 vs. control.

**Figure 5 toxics-11-00021-f005:**
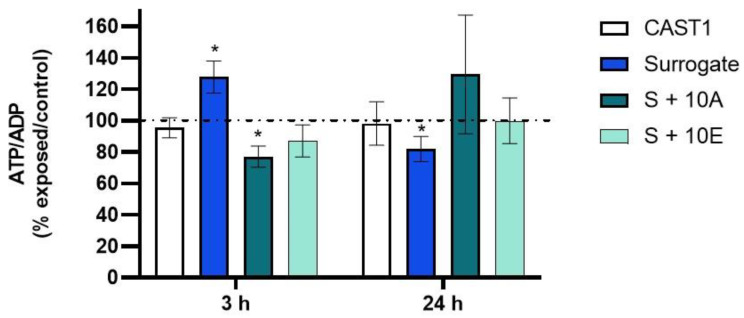
ATP/ADP ratios (in the percentage of air control) in BEAS-2B cells after 3 h or 24 h of exposure to particles or to air. Data are expressed as mean ± SD, n = 4–5 independent experiments. * *p* < 0.05 vs. control.

**Figure 6 toxics-11-00021-f006:**
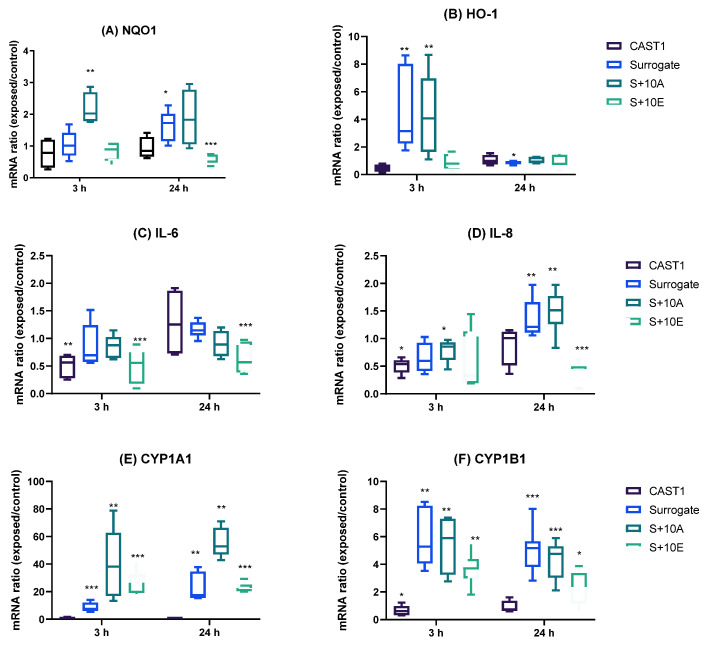
mRNA expression in ALI BEAS-2B cells after 3 h and 24 h exposure. (**A**,**B**): Oxidative stress markers *NQO1* and *HO-1* transcript production. (**C**,**D**): Inflammation markers *IL-8* and *IL-6* transcript production. (**E**,**F**) Xenobiotic metabolism markers *CYP1A1* and *CYP1B1* transcript production. Results are expressed as the median ± SD of at least four independent experiments. Statistical analysis was performed using the non-parametric Mann–Whitney U test (GraphPad for Windows, v9.4.0). Statistically significant differences were reported with * *p* < 0.05, ** *p* < 0.01, *** *p* < 0.001.

**Table 1 toxics-11-00021-t001:** Flow rates of the operating conditions studied.

	CAST1	Surrogate	S+10A	S+10E
Propane (mL/min)	60	0	0	0
Liquid fuel (mL/min)	0	0.3	0.3	0.3
Nitrogen (mL/min)	7	7	7	7
Oxidation air (L/min)	1.5	1.52	1.46	1.5
Dilution air (L/min)	20	20	20	20

**Table 2 toxics-11-00021-t002:** Size distribution and morphological parameters of the four studied aerosols. (A) CAST1; (B) gasoline Surrogate; (C) Surrogate anisole (S+A); and (D) Surrogate ethanol (S+E). * Values from [36].

	CAST1 *	Surrogate	S+10A	S+10E
D_m, geo_ (nm)	114	124	123	123
σ_geo, Dm_	1.54	1.54	1.53	1.53
D_p, geo_ (nm)	26.6	43.3	35.5	34.9
σ_p, geo_	1.31	1.34	1.39	1.45
OC/TC (%)	4.1 ± 3.5	3.5 ± 0.7	5.0 ± 1.1	7.7 ± 0.9

**Table 3 toxics-11-00021-t003:** ATP and ADP levels (nmol/mg protein) in BEAS-2B cells after 3 h or 24 h of exposure to particles or to air (control). Data are expressed as mean ± SD, n = 4–5 independent experiments. * *p* < 0.05 vs. control.

	ATP		ADP	
	3 h	24 h	3 h	24 h
Control	23.60 ± 2.90	29.30 ± 6.29	2.07 ± 0.46	2.02 ± 0.68
CAST1	23.65 ± 2.74	23.70 ± 2.94	1.79 ± 0.61	2.64 ± 1.31
Control	37.40 ± 2.66	38.62 ± 0.99	3.10 ± 0.15	4.14 ± 0.57
Surrogate	38.05 ± 1.36	37.40 ± 6.60	2.49 ± 0.33	4.84 ± 0.39
Control	32.52 ± 5.19	31.07 ± 2.04	1.87 ± 0.44	4.16 ± 0.80
S+10A	29.65 ± 6.96	26.90 ± 4.72	2.19 ± 0.42	2.89 ± 0.69 *
Control	31.48 ± 5.05	18.87 ± 2.03	1.97 ± 0.74	1.91 ± 0.32
S+10E	22.27 ± 4.06	18.26 ± 5.02	1.69 ± 0.11	2.90 ± 0.67

## Data Availability

Not applicable.

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
