# Peer review of "Ultrafine Particles Issued from Gasoline-Fuels and Biofuel Surrogates Combustion: A Comparative Study of the Physicochemical and In Vitro Toxicological Effects"

_toxics, 2022, doi:10.3390/toxics11010021_

Round 1
Reviewer 1 Report
This is a very interesting manuscript. However, my main concern is the poor discussion of the results described. I consider that authors should explain better (and justify) some of their methods and findings, and provide hypothesis that could explain the results found. Some sections are repetitions, and entire paragraphs provided in the discussion should be removed and/or changed. Please consider my comments below:
INTRODUCTION
Authors should consider re-writing the aims and remove things related with material/methods.
MATERIALS AND METHODS
Line 162 – please explain how/where these diesel particles were obtained.
Line 169 – explain why 7 different concentrations were used. Is there any ref to justify this methodology?
Please consider providing references for the two equations provided, i.e. for the %DTT and %AO depletion. Also add references for the qPCR method performed.
Line 247 – explain why those expression profiles were chosen. Explain briefly how normalisation to B2M was performed.
I think that Fig 1 should be better explained.
RESULTS
Line 262 – add the value of the reduction observed.
Figure 3 – is there any statistical differences between the different particles and between the four studied aerosols?
Fig 4 – consider adding the control to the figure provided. Same for Figs 5 and 6.
Explain the times selected of 3 h and 24 h. Is there any justification from the literature?
Consider adding the number of repeats of the experiments also in the MM section.
Re-order the tables and introduce Table 1 (section 3.3) in the Results sections. Describe the statistical differences found.
DISCUSSION
First paragraph is not discussion, it should be removed as it is a repetition. Second paragraph should be moved to the MM section to justify the methods used. Check the other paragraphs and delete any information that it is not a discussion. There are a few repetitions, for example line 526.
Results should be better explained and justified. Hypothesis should be presented that could explain the results.
Line 473 – which metals? Please could you elaborate this more?
Reviewer 2 Report
The work presented by Juárez-Facio et al. characterizes several biofuels and their emissions. Appropriate characterization of the physicochemical characteristics, oxidative potential, and biological effects, which results relevant to the toxicology assessment of air pollutants. However, some questions arise;
1. Page 7, last paragraph states " The relative intensity by molecular classes, presented in Figure 2.a for the four analyzed samples, showed slight differences between combustion conditions and fuels. For all fuels and conditions, the ion relative intensity was dominated by the hydrocarbon class (HC) (around 80%), followed by compounds with one and two oxygen atoms (around 17% and 3 %, respectively). However, CAST1 showed a higher relative intensity for the hydrocarbon molecular class and a lower relative intensity for the oxygenated molecular classes." The performed analysis to establish these differences, the p values, and the corresponding marks in the figure should be added
2. The statistical analysis section should describe the PCA and how these values were taken into account for the Figures 2c and 2d
3. The evaluated genes are representative of different cellular processes, metabolism, antioxidant defense, and inflammation, however, is not clear why the authors selected them and not a different set of genes i.e, nrf2, SODs, catalase, glutathione peroxidases for antioxidant defense and for inflammation TNF alpha or IL 1B
4. The authors state the study limitations in a proper way, as they mentioned in a real scenario, the analyzed PM characteristics may change, can the authors expand on this in the discussion section?
Minor
1. CHO needs to be spelled out the first time is used
2. Table 1, the third column header should be ADP instead of ATP
Round 2
Reviewer 1 Report
Thank you for addressing my comments. I believe that the revised manuscript is stronger and recommend its publication.